# Experience and satisfaction towards palliative care in an Ethiopian tertiary care setting: A mixed methods study of patients with cancer and caregivers

**Atalay Mulu Fentie**[1]*, **Edom Seife**[2], **Sachiko Ozawa**[3], **Teferi Gedif Fenta**[4]

**1** Department of Pharmacology and Clinical Pharmacy, School of Pharmacy, Addis Ababa University, Addis Ababa, Ethiopia, **2** Oncology and Radiotherapy Department, School of Medicine, Addis Ababa University, Addis Ababa, Ethiopia, **3** Eshelman School of Pharmacy, the University of North Carolina at Chapel Hill, Chapel Hill, North Carolina, United States of America, **4** Department of Social and Administrative Pharmacy, School of Pharmacy, Addis Ababa University, Addis Ababa, Ethiopia

* atalay.mulu@aau.edu.et

## Abstract

Patients and caregivers often face significant challenges in addressing palliative care needs in Ethiopia. This study aimed to assess patients treated for cancer and family caregiver satisfaction and experience towards palliative care at Tikur Anbessa Specialized Hospital (TASH). An explanatory mixed method study was conducted among patients treated for cancer and their family caregivers at TASH and data was collected from May 13 to August 30, 2024. A survey of 844 patients with cancer to assess satisfaction using SATMED-Q was conducted followed by interviews of 11 patients and 8 caregivers to explore their experience towards palliative care. Descriptive statistics summarized patient characteristics and a multiple linear regression analysis was conducted to identify factors associated with overall satisfaction. Qualitative analysis utilized thematic content analysis using MAXQDA. Overall mean patient satisfaction for palliative care was 56.7%±3.86. Significantly lower satisfaction was found among patients residing outside of Addis Ababa (β: -2.77, 95%CI:-5.29, -0.25, $p$:0.031). Lower satisfaction was also observed among patients with the reproductive system (β: -6.64, 95%CI:-10.38, -2.91, $p<0.0001$), gastrointestinal (β: -4.11, 95%CI: -7.85, -0.38 $p$:0.031), head and neck (β: -11.56, 95%CI: -16.63, -6.50, $p<0.001$), and lung and mediastinum (β: -9.96, 95%CI: -16.63, -6.50, $p<0.001$) cancers as compared to patients with breast cancer. Patients experiencing moderate (β: 95%: -7.34, 95%CI: -10.26, -4.42, $p<0.001$) and severe pain (β:-11.5, 95%CI: -13.05, -7.98, $p<0.001$) also had lower satisfaction. Lack of whole-patient care, financial constraints, access to pain medications, physical inaccessibility of the care, poor coordination and continuity of care, and hospital infrastructure were mentioned as major systemic barriers. The study reveals significant dissatisfaction with palliative care in Ethiopia among patients with cancer, primarily due to pain management challenges,

**Data availability statement:** All relevant data are within the paper and its Supporting information files.

**Funding:** This study was financially supported by Pfizer in the form of a Pfizer Global Medical Grant awarded to and administered by Addis Abbaba University to partially support the salaries of AMF and ES (76384195). This study was further financially supported by Addis Ababa University in the form of an Addis Ababa University Thematic Research Grant awarded to AMF and ES to support the data collectors' costs (RD/LT-305/2022). No additional external funding was received for this study. The funders had no role in study design, data collection and analysis, decision to publish, or preparation of the manuscript.

**Competing interests:** The authors have declared that no competing interests exist.

financial barriers, and systemic inefficiencies. These findings are consistent with previous studies and underscore the urgent need to integrate palliative care into primary healthcare, improve access to pain medications and coordinate holistic care.

## Introduction

Palliative care, a specialized form of care that plays a crucial role in managing physical, emotional, and psychological distress, can improve the quality of life for patients with serious, life-limiting illnesses [1]. Globally, the experience and satisfaction of patients with cancer and family caregivers in receiving palliative care are shaped by numerous factors, such as the availability of essential medications, the quality of care, the responsiveness and training of healthcare professionals, and the healthcare infrastructure [2–4]. Studies showed patient-centered care significantly contributed to higher satisfaction levels among patients with cancer and caregivers [3,5]. However, despite advancements in palliative care services, there remain significant disparities in access and quality of care across different regions. For example, the wealthiest 10% of the global population consume 90% of the world's opioids, while the poorest 50% has access to only 1% [6]. Additionally, over 90% of certified hospice and palliative physicians and nurse practitioners globally are located in metropolitan areas [7].

The experience of palliative care in oncology in high-income countries is often characterized by a well-coordinated, patient-centered approach. Palliative care services in these settings often includes pain management, mental health, and spiritual care within an accessible and integrated healthcare system [7,8]. However, in low- and middle-income countries (LMICs) such as Ethiopia, palliative care services are less established, mostly not holistic, and frequently face significant challenges. Some of the challenges includes limited access to care, shortage of trained professionals, lack of dedicated units, poor coordination, inadequate home-based palliative services, resource constraints, and financial barriers, leading to more financial, emotional and psychological difficulties to patients and families [7,9,10].

In high-income countries, where healthcare infrastructure and home-based palliative care services are well established, patient satisfaction is generally higher due to better access to essential medications and comprehensive care [7,8,11]. In contrast, in LMICs like Ethiopia, satisfaction with palliative care among patients with cancer is often hindered by the unavailability of essential medicines such as morphine, high out-of-pocket costs, and limited service integration [9,10]. Cultural and religious values also shape perceptions of care; for example, Ethiopian patients may express spiritual gratitude despite unmet clinical needs, potentially masking dissatisfaction [12,13].

Globally, studies have shown that integrated and home-based palliative care models are associated with higher satisfaction. For instance, satisfaction improved significantly when palliative care was integrated into geriatrics services in Sweden [14] and satisfaction with home-based palliative care in the Bangladesh study was 88.2% [15]. However, in Ethiopia, no published study has

specifically assessed patient or caregiver satisfaction with palliative care among patients diagnosed and treated for cancer. There was a single knowledge, attitudes, and practices (KAP) study among cervical cancer patients and found that only 26% could define palliative care, though 80.5% had a positive attitude [16]. The rest KAP studies were on among healthcare professionals, particularly nurses, with the pooled prevalence of palliative care good knowledge of (42.31%) [17].

Ethiopia has formally recognized palliative care within national health strategies and has endorsed palliative care guideline, a five years strategic plan 2025–2029 and included pain as the 5th vital sign. Despite oral morphine also being included in the national Essential Medicines List, access to opioid analgesics remains extremely limited, and consumption is among the lowest globally, at <3mg/person/year, compared with a global average of approximately 32mg/person/year. This persistent disparity reflects regulatory, supply chain, and health system constraints and contributes to substantial under-treatment of pain among palliative care population [6,10]. Moreover, palliative care in Ethiopia is generally constrained by limited multidisciplinary team involvement, with services largely delivered by physicians and nurses and minimal integration of psychosocial or spiritual care. Hospice Ethiopia is providing home-based palliative care in Addis Ababa since 2003 and more recently there are initiatives that integrated hospital–home-based care model and gradually expanding [10].

Therefore, all these gaps underscore the need for exploring patients' with cancer and family caregivers' experiences and satisfaction with palliative care in the Ethiopian context. Globally, there is growing recognition of the importance of improving palliative care services to provide compassionate, comprehensive care for those facing life-threatening illnesses [6,14,15]. Our study addresses this by employing a mixed-methods approach to understand how palliative care is perceived and experienced. Hence, this study aimed to explore patients' with cancer and family caregivers' experience and understand patient satisfaction with palliative care by employing both quantitative and qualitative methods. The findings aim to inform policies and service design to ensure accessible, holistic, and culturally sensitive palliative care in Ethiopia and similar settings.

## Methods

### Ethics statement

Our study was conducted in accordance with the principles of the Declaration of Helsinki. Ethical approval for the study was granted by the Institutional Review Board of the College of Health Sciences, at Addis Ababa University (**protocol number 054/23/SoP**). Informed consent, both oral and written, was obtained from all participants, who were made aware that participation was voluntary and that they could withdraw at any time without any repercussion. The entire consent procedure, including the approach for verbal consent, was reviewed and approved by the ethics committee. No monetary incentives were provided, and participation did not affect their care. Data were securely stored, anonymized, and analyzed in aggregate to ensure confidentiality.

### Study setting

Tikur Anbessa Specialized Hospital (TASH) is a pioneer oncology center that provides comprehensive cancer care for the entire nation through its emergency, inpatient, and outpatient services. In accordance with the WHO definition of palliative care [1], supportive and palliative care are intended to be integrated from the time of cancer diagnosis, regardless of whether patients are receiving curative, adjuvant, or disease-modifying treatments. In this setting, palliative care focuses on early symptom assessment and management; particularly pain control (pain is the 5th vital sign in the hospital), which is the most prevalent symptom along with psychological, and social support as needed. Even if the awareness, understanding and level of palliative care service varies from unit to unit and by professionals, these services are primarily delivered within routine oncology care pathways. As a result, patients receiving curative treatments may still receive elements

of palliative care, including symptom relief, counseling, and other supportive interventions, depending on their clinical needs. In addition, short-stay intermediary inpatient wards are designated for palliative care services. Within the oncology department, a dedicated outpatient palliative care clinic operates for two half-days each week (Tuesday and Thursday) to provide palliative care consultations and address other patient concerns.

## Research design

This study employed explanatory mixed methods, where a quantitative survey was followed by qualitative research techniques, and data was collected from May 13 to August 30, 2024. The quantitative phase utilized the Treatment Satisfaction with Medicines questionnaire (SATMED-Q), a standardized tool designed to assess various dimensions of patient satisfaction, providing numerical data that quantifies overall satisfaction. Subsequently, the qualitative phase involved in-depth interviews with patients to capture rich, detailed insights into their experiences, perceptions, and factors influencing their satisfaction that may not be fully captured through standardized scales [18,19]. A mixed methods approach was chosen to allow for a comprehensive understanding of patient satisfaction and experience, combining the broad, generalizable data from the quantitative phase with the nuanced, context-specific insights from the qualitative phase. This combination provides a more complete and holistic picture of patient satisfaction towards palliative care and their experiences (Fig 1).

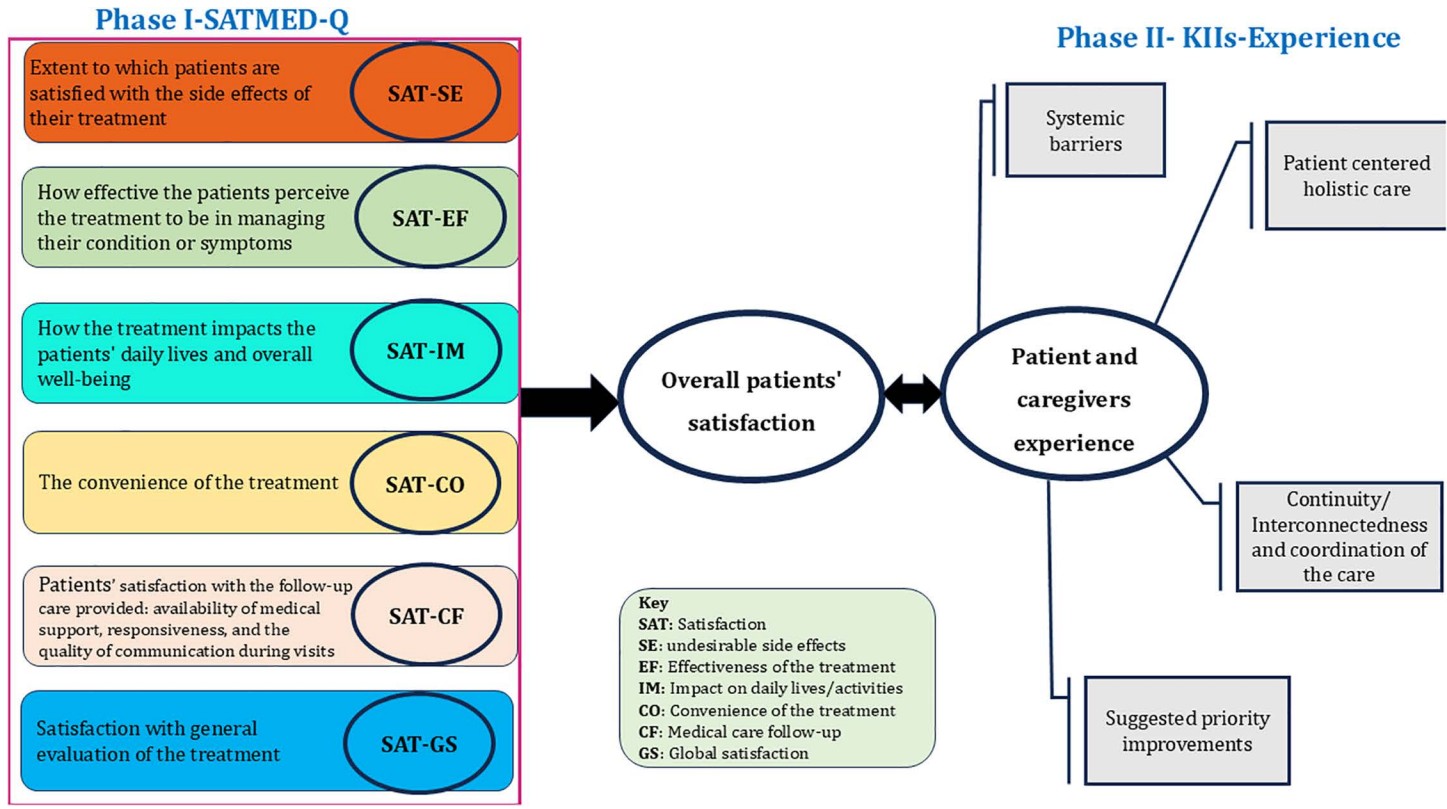

**Fig 1. Conceptual framework of patient satisfaction and family-caregivers experience.**

## Participant recruitment

The study participants for the quantitative survey were all patients with confirmed cancer diagnosis treated at TASH. Those patients who were willing to participate, age ≥ 18 years, who were able to communicate effectively and available during the study period were surveyed using a consecutive sampling technique. Since no similar study had been conducted in the country to provide an estimate of the satisfaction level, we used a proportion of 50% to obtain the maximum possible sample size. The sample size was calculated using the single population proportion formula [20], with a 95% confidence level, 0.05 margin of error, and giving an initial estimate of 384 participants. Accounting for 10% non-responses rate and inappropriate responses as well as a design effect of 2 to account for the multistage sampling approach (initially at department level and the outpatient and inpatient level), the study finally recruited a total of 844 patients.

For the qualitative in-depth key informant interviews, a maximum variation purposive sampling technique was used to recruit patient participants with a confirmed cancer diagnosis and their respective family caregivers. Participants were selected based on their patient follow-up status and the primary site of cancer, continuing until data saturation was reached. In total, 19 participants (11 patients and 8 family caregivers) were recruited for the study.

## Data collection techniques

For the quantitative survey, demographic characteristics were obtained either directly from the patient or caregiver, or from the electronic medical record (EMR) using a standardized checklist. Clinical and treatment-related characteristics were collected from the EMR using a pre-prepared checklist, following the acquisition of written informed consent and the respective patient's card/EMR number. Data on patient satisfaction with palliative care services were gathered through face-to-face interviews conducted directly with the patients using the Amharic version of the SATMED-Q, which was validated and widely used in Ethiopia and beyond [21,22]. This tool is designed to evaluate patients' satisfaction with their medication treatment across multiple dimensions. The questionnaire consists of 17 items, which are categorized into 6 subdomains. Each subdomain contains specific items related to different aspects of treatment satisfaction: Undesired Side Effects (Items 1–3) (**SAT-SE**); Effectiveness (Items 4–6) (**SAT-EF**); Impact (Items 7–9) (**SAT-IM**); Convenience (Items 10–12) (**SAT-CO**); Medical Care Follow-up (Items 13–14) (**SAT-CF**); and Global Satisfaction (Item 15–17) **(SAT-GS)**. Patients were asked to rate their level of satisfaction with each item on a scale from 0 to 4, where: 0 = Not at all; 1 = A little bit; 2 = Somewhat; 3 = Quite a bit; 4 = Very much. Presence of pain and average pain severity during last week was measured using a visual analogue scale from 0 (no pain) to 10 (worst pain).

In-depth key informant interviews were conducted using a standardized interview guide (S1 Appendix). The guide included various probing questions designed to facilitate a deep understanding of their experience towards palliative care. It focused on exploring patients' and family caregivers' real-time experiences, specifically addressing systemic barriers, the interconnectedness of care, patient-centered care, holistic care, and the continuity and coordination of care. During the interviews, clear instructions were provided to ensure a supportive, trusting, and empathetic environment, encouraging participants to share their genuine experiences without fear of judgment or repercussions. This approach aimed to foster open and honest communication, ensuring the participants felt comfortable throughout the process (Fig 1) [23].

Both SATMED-Q and interview guide were translated to Amharic, the commonly spoken and national language of Ethiopia and back translated independently by English and Amharic language fluent oncologists.

## Data analysis

Descriptive statistics was used to summarize demographic, clinical and treatment characteristics using measures of dispersion and central tendency. Kolmogorov-Smirnov test was applied to test normality. The SATMED-Q overall composite direct score ranged from 0 to 68, and was transformed to 0 (no satisfaction) to 100 (maximum expected best satisfaction)

for ease of understanding and interpretation using the below formula. Similarly, the same formula was used to transform it to 0–100 per respective subdomains' total direct score.

$$X = [(X_{obs} - X_{min})/(X_{max} - X_{min})] \times 100$$

where $X_{max}$ is the maximum total score for the overall and subdomains, $X_{min}$ is the minimum total score for the overall and subdomains, $X_{obs}$ is the total patient score for the overall and each subdomain, and $X$ is the transformed score out of 100. One way analysis of variance (ANOVA) was used to observe the relationship between the satisfaction outcome variable and independent categorical variables. Following ANOVA, all variables that had a $p$-value <0.2 in the bivariate analysis were included in the final multivariable linear regression model and $p$-value <0.05 was considered statistically significant. All data analysis was performed using SPSS version 27 (IBM Corp., Armonk, NY, USA).

All key informant interviews were recorded after obtaining participants' written consent, transcribed verbatim, and translated. The data collected from the interviews were organized using MAXQDA V24.1 software and analyzed using both deductive and inductive coding where there were pre-set themes/aims or objectives as well as new themes or sub-themes that emerged out of the data. Line-by-line coding, writing of memos, indexing, charting and data mapping were conducted, and important local terms related to palliative care was developed into a glossary (የስቃይ ማስታገሻ፤ የስቃይ እፎይታ ፤ እረፍት የሚያስገኝ ህክምና). Quotes corresponding to each code were reviewed, and matrices and memos were utilized to organize and identify themes.

## Data quality assurance

The overall reliability or internal consistency of SATMED-Q was checked and gave Cronbach-α value of 0.84, indicating a high level of internal consistency of the items, where the domain specific internal consistency ranged from 0.829 for global satisfaction to 0.971 for satisfaction of treatment effectiveness. The item level Cronbach-α ranged from 0.814 to 0.860 (Table A in S2 Appendix). Discriminant validity was assessed using the pain severity status using the visual analogue scale (VAS) scores (≤mild, moderate, and severe). The analysis revealed that patients with severe pain reported significantly low satisfaction ($p$-value<0.001), indicating a very strong association between lower satisfaction and pain severity status. This finding supports the discriminant validity of the SATMED-Q, demonstrating that the tool effectively differentiates between groups with different levels of pain and treatment satisfaction (Table B in S2 Appendix). To ensure rigor, reliability, and validity, the research question and methodology were developed in consultation with senior qualitative research experts (SO, TGF) and a palliative care specialist (ES). Member checking was conducted with two patients on palliative care to confirm that the interpretations aligned with patients lived experiences.

The first author (AMF) having over a decade of oncology experience in Ethiopia, where palliative care needs are substantial, practiced reflexivity throughout the study to engage a diverse group of patient participants. His positionality provided both insights and limitations, and he remained mindful of how his background could influence participants' responses and the interpretation of data.

## Results

### Patients' characteristics

About 879 eligible patients were approached for the study and a total of 264 inpatients and 580 outpatients were complete the survey giving a response rate of 96.1%. The majority of participants were female (67.9%), with a median age of 49 years. A significant proportion of participants were married (67.4%), and a slightly higher proportion were unable to read or write (27.6%). The majority of participants had breast cancer (22.7%), reproductive system cancer (21.6%), or gastrointestinal tract cancer (21.4%). Regarding performance status, most participants were able to function well, with an ECOG score of ≤2 (89.1%). Approximately 21.9% of participants had at least one comorbid condition, with hypertension (27.3%)

and HIV/AIDS (22.4%) being the most common. Most patients (59.4%) reported that their cancer care was primarily funded through the community-based health insurance system (Table 1).

## Medication prescribing pattern

At the time of the study, nearly half of the patients (n = 408) received treatment to manage their physical symptoms. The most common treatments included pain relief medications, followed by antiemetics and chemotherapy. The most frequently prescribed medications were Tramadol (32.1%), followed by ondansetron (28.9%), metoclopramide (27.5%), and dexamethasone (23.0%). Although 17.7% of patients reported severe pain, only nine (2.2%) were able to obtain morphine (Table 2).

## Patient satisfaction level

The overall mean ± SD composite measure of patient's satisfaction towards palliative care received at TASH was 56.7% ± 3.89. Among SATMED-Q satisfaction sub-domains, the lowest mean ± SD satisfaction was reported for undesirable side effects subdomain (53.3% ± 6.13) and the highest towards global satisfaction (72.3% ± 4.72) subdomain. Mean ± SD satisfaction towards treatment effectiveness, care impact on daily living/activities, convenience, and medical care follow-up subdomains were 54.6% ± 4.65, 56.2% ± 5.67, 61.3% ± 5.32, and 69.1% ± 4.89, respectively.

Statistically significant decreases in the overall SATMED-Q satisfaction scores were observed in the following groups: patients residing outside Addis Ababa, ($p < 0.001$), and among employed patients ($p = 0.045$). Furthermore, patients treated for various malignancies, including reproductive system/gynecological, head and neck, musculoskeletal system ($p < 0.001$), as well as lung and mediastinal ($p = 0.002$) and central nervous system malignancies ($p = 0.03$), reported significantly lower satisfaction scores when compared to patients with breast cancer. Newly diagnosed patients ($p = 0.04$) and those experiencing moderate or severe pain ($p < 0.001$) were found to have lower levels of overall satisfaction with their treatment and care (Table 3).

## Factors associated with overall satisfaction towards palliative care

A multiple linear regression model was run predicting overall patient satisfaction controlling for various variables. Of the six variables that met the inclusion criteria for the final regression model, residence, cancer site, and pain severity at the time of the survey were significantly associated with overall patient satisfaction. The model explained 57.9% of the variance ($p < 0.001$), indicating moderate fit. All independent variables had variance inflation factors (VIF) close to 1, and the tolerance values (1/VIF) ranged from 0.620 to 0.974, suggesting no multicollinearity issues.

Keeping other predictors constant, patients who came from outside Addis Ababa a significantly lower satisfaction (β = –2.77, 95% CI: –5.29, –0.25; $p = 0.03$) as compared to patients living in Addis Ababa where the treatment center is located. Similarly, patients with cancer of the reproductive system (β = -6.64, 95%CI: -10.38, -2.91, $p < 0001$), GIT (β = -4.11, 95%CI:-7.85, -0.38, $p = 0.031$), head and neck (β = -11.56, 95%CI: -16.63, -6.50, $p < 0.001$), lung or mediastinum (β = -9.96, 95%CI: -16.63, -6.50, $p < 0.001$), genitourinary system (β = -7.95, 95%CI: -14.50, -1.40, $p = 0.017$), musculoskeletal system (β = -12.15, 95%CI:-18.02, -6.27, $p < 0.001$) and central nervous system (β = -17.83, 95%CI: -32.54, -3.14, $p = 0.018$) reported a substantially lower satisfaction scores when compared with patients with breast cancer.

Additionally, patients experiencing moderate or severe pain had markedly lower satisfaction levels, with mean scores reduced by 7.34 percentage points ($p < 0.001$) and 11.5 percentage points ($p < 0.001$), respectively, compared to those with mild or no pain (Table 4).

## Perspectives of patients/family caregivers towards palliative care

The qualitative analysis generated four interrelated themes; (1) overall care experience and satisfaction, (2) systemic barriers, (3) interconnectedness and continuity of care, and (4) patient-centered holistic care, which help triangulate and

**Table 1. Democratic characteristics of studied patients (N = 844).**

| Variable, N = 844 | | N(%) |
|---|---|---|
| **Sex** | Female | 573(67.9) |
| Age in years | Median(IQR) | 49 (38-58) |
| Residence | Out of Addis Ababa where the hospital is located | 456(54.0) |
| Current marital status | Never married | 101(12.0) |
| | Married | 569(67.4) |
| | Divorced | 82(9.70) |
| | Widowed/er | 92(10.9) |
| Educational status | Can't read and write | 233(27.6) |
| | Can read and write without formal education | 50(5.9) |
| | Primary education | 218(25.8) |
| | Secondary education | 213(25.2) |
| | Diploma and above | 130(15.4) |
| Employment | Unemployed* | 430(50.9) |
| | Employed** | 414(49.1) |
| Primary site of cancer | Breast | 192(22.7) |
| | Reproductive system | 182(21.6) |
| | Gastrointestinal tract (lower or upper) | 181(21.4) |
| | Blood and bone marrow | 97(11.5) |
| | Head and neck | 71(8.4) |
| | Musculoskeletal system | 45(5.3) |
| | Lung and mediastinum | 36(4.3) |
| | Urinary system | 34(4.0) |
| | Central nervous system | 6(0.7) |
| Type of diagnosis | New | 686(81.3) |
| | Relapse-recurrence | 158(18.7) |
| Performance status | Able to function (ECOG 0–1) | 752(89.1) |
| | Not able to function (ECOG 2–4) | 92(10.9) |
| Presence of comorbidities | Yes | 183(21.9) |
| Type of comorbidity, n = 183 | Hypertension | 50(27.3) |
| | HIV/AIDS | 41(22.4) |
| | Hypertension + diabetes | 28(15.3) |
| | Diabetes mellitus | 16(8.7) |
| | Asthma | 12(6.6) |
| | Others*** | 36(19.7) |
| Goal of treatment for those started during the survey = 714 | Curative | 497(69.6) |
| | Palliative | 217(30.4) |
| Source of payment for the cancer care | Community based health insurance | 501(59.4) |
| | Out of pocket | 308(36.5) |
| | Covered by the employing company | 35(4.1) |

*Includes anyone with a job, whether full-time, part-time, or self-employed; **All individuals who are not currently working, which may include the unemployed, retired, students, or those not in the labor force for other reasons. *** CKD = 9; CHF = 6; hyperthyroidism = 5; chronic hepatitis = 4; DVT = 4; hypothyroidism = 3; Epilepsy = 3; Atherosclerosis = 2.

**Table 2. Palliative care medications prescribing pattern to treat physical symptoms.**

| Variable, N=844 | | N(%) |
|---|---|---|
| Pain severity at time of study using Visual Analogue Scale | Mild (≤3) | 473(56.0) |
| | Moderate (4–6) | 222(26.3) |
| | Severe (7–10) | 149(17.7) |
| Patient currently taking any documented palliative care drug treatment | Yes | 408(48.3) |
| Type of palliative care used during the survey, n=408 | Anti-pain medications alone | 164(40.2) |
| | Antiemetic medications alone | 86(21.1) |
| | Chemotherapy + antiemetic | 49(12.0) |
| | Gastric acid suppressing agents | 25(6.1) |
| | Antipain + antiemetic + chemotherapy | 24 (5.9) |
| | Radiotherapy and/or skin care/oral care | 20(4.9) |
| | Antipain plus laxatives | 18 (4.4) |
| | Laxative alone | 11 (2.7) |
| | Others* | 11(4.0) |
| Prescribed medications for palliative care other than chemotherapy | Tramadol | 131 (32.1) |
| | Ondansetron | 118 (28.9) |
| | Metoclopramide | 112(27.5) |
| | Dexamethasone | 94(23.0) |
| | Omeprazole | 33(8.1) |
| | Diclofenac | 29(7.1) |
| | Ibuprofen | 25(6.1) |
| | Bisacodyl | 23(5.6) |
| | Paracetamol | 19(4.7) |
| | Cimetidine | 16(3.9) |
| | Pregabalin | 12(2.9) |
| | Zoledronic acid | 11(2.7) |
| | Morphine | 9(2.2) |
| | Amitriptyline | 8(2.0) |
| | Others** | 8(2.0) |

*Antipain plus gastric acid suppressing agents=6; chemotherapy + mineral supplements=2; Radiotherapy + antipain + antiemetic=2; **Meloxicam=5, Indomethacin=3.

contextualize the quantitative findings on satisfaction with palliative care at TASH. Together, these themes provide explanatory depth and highlight system-level and care-process factors influencing patient and family caregiver experiences.

Key informant interview participants generally described their experience and satisfaction with palliative care at TASH as modest, reflecting a mix of appreciation and concern. Some expressed gratitude for the care received, with one patient noting:

*"Thank God, I have no complaints." (P010)*

However, several participants highlighted significant challenges that negatively affected their care experience. the hospital's physical infrastructure was described as inadequate for the needs of seriously ill patients, particularly in terms of basic facilities:

*"The hospital lacks basic amenities such as a functioning toilet." (F/C006)*

 

**Table 3. Relationship between patient characteristics and satisfaction.**

| Variables | | SATMED-Q Domains out 100, Mean±SD | | | | | | |
|---|---|---|---|---|---|---|---|---|
| | | SAT-Overall | SAT-SE | SAT-EF | SAT-CO | SAT-IM | SAT-CF | SAT-GS |
| Residence | AA | 59.6±7.81 | 53.2±6.90) | 56.1±8.22 | 64.5±4.56) | 59.9±6.23 | 73.3±3.72 | 75.9±5.65 |
| | Out of AA | 54.2±4.93 | 53.5±5.99 | 53.3±3.99 | 58.5±9.11 | 53.1±4.77 | 65.6±8.12 | 69.1±8.43 |
| | *p*-value | *<0.001* | *0.911* | *0.216* | *0.005* | *0.002* | *<0.0001* | *<0.0001* |
| Employment | Unemployed | 57.9±3.99 | 56.0±6.44 | 56.2±4.73 | 62.3±7.22 | 56.7±6.22 | 70.7±9.11 | 72.6±5.33 |
| | Employed | 55.3±5.89 | 50.3±10.23 | 52.9±7.85 | 60.2±5.85 | 55.7±4.33 | 67.5±6.65 | 71.9±7.23 |
| | *p*-value | *0.045* | *0.019* | *0.145* | *0.341* | *0.670* | *0.137* | *0.665* |
| Site of cancer | Breast | 63.7±6.64 | 55.5±7.33 | 62.6±5.66 | 68.1±8.88 | 68.1±6.43 | 74.6±7.33 | 76.2±7.88 |
| | Reproductive system | 54.0±5.40 | 55.0±9.23 | 52.2±9.67 | 57.2 ± (6.94 | 52.6±6.42 | 66.6±8.11 | 69.1±5.99 |
| | GIT (lower or upper) | 56.5±6.80) | 54.5±12.11 | 54.7±11.23 | 57.3±6.43 | 55.9±13.12 | 70.0±8.23 | 73.3±7.88 |
| | Head and neck | 48.4±8.21 | 47.1±13.58 | 47.4±10.11 | 49.3±11.23 | 46.9±10.00 | 60.0±6.77 | 67.1±8.32 |
| | Lung & mediastinum | 50.2±7.66) | 54.7±9.66 | 38.2±9.45 | 58.3±12.11 | 46.3±11.30 | 70.5±8.65 | 70.1±8.55 |
| | Blood & bone marrow | 61.3±4.76 | 51.4±8.43 | 61.3±7.88 | 68.8±6.33 | 61.2±5.78 | 70.6 ± (6.49 | 76.7±6.32 |
| | Genito-urinary system | 54.1±6.90 | 50.9±14.21 | 48.8±12.11 | 59.6±10.28 | 50.0±9.99 | 77.6±8.45 | 70.8±9.14 |
| | MSS | 48.8±11.26 | 43.9±11.2 | 43.7±10.32 | 55.6±9.32 | 43.5±10.43 | 56.7±12.31 | 66.8±8.76 |
| | CNS | 54.1±13.23) | 75.6±6.43 | 41.7±14.51 | 25.0±13.21 | 16.7±9.65 | 60.4±13.55 | 59.7±12.11 |
| | *p*-value | *<0.0001* | *0.149* | *<0.0001* | *<0.0001* | *<0.0001* | *0.006* | *0.001* |
| Performance status | Able to function | 56.9±3.57 | 52.1±5.34 | 54.5±4.09 | 62.2±5.44 | 57.1±6.03 | 69.6±5.44 | 72.9±3.09 |
| | Not able to function | 54.7±6.55 | 62.0±7.53 | 55.1±7.43 | 53.5±6.45 | 48.9±5.65 | 65.5±6.34 | 67.1±6.77 |
| | *p*-value | *0.300* | *0.007* | *0.874* | *0.013* | *0.20* | *0.250* | *0.014* |
| Diagnosis type | New | 54.1±4.99 | 54.4±6.34 | 54.1±3.76 | 60.5±3.67 | 61.8±5.45 | 69.1±5.76 | 72.4±5.65 |
| | Relapse/recurrence | 60.1±5.45 | 49.5±6.77 | 56.8±6.45 | 64.4±6.45 | 54.9±4.65 | 69.2±5.66 | 71.7±6.55 |
| | *p*-value | *0.04* | *0.101* | *0.346* | *0.164* | *0.014* | *0.964* | *0.727* |
| Pain severity | ≤Mild | 61.2±3.46 | 49.5±5.34 | 61.2±5.43 | 65.9±4.55 | 64.3±6.55 | 74.8±5.32 | 76.4±3.45 |
| | Moderate | 52.6±4.32 | 52.9±6.34 | 47.4±7.34 | 60.4±5.34 | 49.0±6.33 | 64.9±4.09 | 69.9±5.43 |
| | Severe | 48.5±5.76 | 65.3±6.77 | 44.5±5.34 | 47.8±6.77 | 41.8±6.45 | 57.3±6.32 | 62.5±5.34 |
| | *p*-value | *<0.0001* | *<0.0001* | *<0.0001* | *<0.0001* | *<0.0001* | *<0.0001* | *<0.0001* |
| Overall composite score out of 100 | | 56.7±3.89 | 53.3±6.13 | 54.6±4.65 | 61.3±5.32 | 56.2±5.67 | 69.1±4.89 | 72.3±4.72) |

**SAT-Overall:** overall satisfaction; **SAT-SE:** Undesirable side-effects on satisfaction; **SAT-EF:** Satisfaction on treatment effectiveness; **SAT-CO:** Satisfaction on convenience; **SAT-IM:** Satisfaction on impact on daily living/activities; **SAT-CF:** Satisfaction on medical care/follow-up; **SAT-GS:** Global satisfaction; **AA:** Addis Ababa (where the facility is located); **GIT:** Gastro-intestinal tract; **MSS:** Musculoskeletal system: **CNS:** Central nervous system.

Another major concern was on the continuity in care due to frequent changes in healthcare providers, which caused frustration and repeated explanations of their medical history:

restricted access to care outside scheduled appointment times, limiting patients' ability to seek timely support: One family caregiver participant said:

*"It is difficult to seek treatment outside the scheduled appointments." (P008, F/C003)*

These diverse perspectives reflect the complexity of patient experiences and highlight critical areas for improvement in service continuity, accessibility, and infrastructure within the palliative care setting at TASH.

## Systemic barriers to palliative care

When patients and caregivers were asked about systemic barriers they faced in accessing palliative care and satisfaction towards palliative care, all participants mentioned financial challenges as the major barrier and significant concern.

**Table 4. Factors associated with overall satisfaction towards palliative care.**

| Variable | | Reference | β-coefficient(95%CI) | S.E | p-value | Collinearity tests | |
|---|---|---|---|---|---|---|---|
| | | | | | | Tolerance | VIF |
| Residence | Out of Addis Ababa | Addis Ababa | -2.77(-5.29, -0.25) | 1.28 | **0.031** | 0.924 | 1.10 |
| Employment status | Employed | Unemployed | -1.99(-4.51, 0.53) | 1.28 | 0.123 | 0.917 | 1.10 |
| Performance status | Able to function | Not able to function | 1.23(-2.85, 5.31) | 2.10 | 0.554 | 0.900 | 1.12 |
| Diagnosis type | New | Relapse/recurrence | 2.91(-0.23, 6.04) | 1.59 | 0.069 | 0.974 | 1.03 |
| Cancer site*, reference: breast cancer | Reproductive system | Breast | -6.64(-10.38, -2.91) | 1.90 | **<0.001** | 0.807 | 1.24 |
| | Gastrointestinal tract | Breast | -4.11(-7.85, -0.38) | 1.90 | **0.031** | 0.620 | 1.61 |
| | Head and neck | Breast | -11.56(-16.63, -6.50) | 2.58 | **<0.001** | 0.621 | 1.61 |
| | Lung and mediastinum | Breast | -9.96(-16.63, -6.50) | 2.58 | **<0.001** | 0.737 | 1.36 |
| | Blood and bone marrow | Breast | -1.37(-5.80, 3.06) | 2.25 | 0.544 | 0.729 | 1.37 |
| | Genitourinary system | Breast | -7.95 (-14.50, -1.40) | 3.34 | **0.017** | 0.878 | 1.14 |
| | Musculoskeletal system | Breast | -12.15(-18.02, -6.27) | 2.99 | **<0.001** | 0.836 | 1.20 |
| | Central nervous system | Breast | -17.83(-32.54, -3.14) | 7.49 | **0.018** | 0.950 | 1.05 |
| Pain*, reference: mild pain | Moderate | ≤Mild | -7.34 (-10.26, -4.42) | 1.49 | **<0.001** | 0.881 | 1.13 |
| | Severe | ≤Mild | -11.5(-13.05, -7.98) | 1.79 | **<0.001** | 0.807 | 1.24 |

*Dummy variable was created and comparison was made against every other cancer site and only those variables that fulfilled the inclusion criteria of p-value<0.2 were included in the final multivariable linear regression model. ***Predicted overall patient satisfaction*** *= 67.39 - 2.77 (out of Addis Ababa) - 6.64 (patients with reproductive system cancer) - 4.11 (patients with GIT cancer) - 11.56 (patients with head and neck cancer) - 9.96 (patients with lung and mediastinum cancer) - 7.95 (patients with genitourinary system cancer) - 12.15 (MSS cancer) - 17.83 (patients with CNS cancer) - 7.34 (patients with moderate pain) - 11.5 (patients with severe pain).*

Particularly, limited access and high cost of medications and diagnostic tests were mentioned frequently. For example, pain-relieving medications such as morphine and necessary diagnostic procedures were either unavailable or prohibitively expensive, creating a substantial financial burden to patients and their families. Those living in rural areas also mentioned inaccessibility of palliative care services to their vicinity as a major challenge. The long travel times to reach TASH placed both physical and financial strain for both patients and families/caregivers (Table 5).

Another major challenge was the long wait times for radiation therapy and chemotherapy, often leading to disease progression and relapse, and reason for patients to be left with limited options to survive without suffering. In addition, both patients and family caregivers pointed to inadequate infrastructure as a barrier to care. Poor hospital conditions, including a lack of proper sanitation and functioning toilets, further complicated their treatment experience. The high patient load and frequent provider changes were also noted as challenges, forcing patients to repeatedly explain their medical history, leading to inconsistent care and communication (Table 5).

## The interconnectedness and continuity of care

The interconnectedness and continuity of palliative care in the interview transcripts reveals significant gaps and challenges in providing comprehensive and coordinated care. While patients receive treatment from various healthcare providers, the care process often lacks continuity and coordination, leading to delays, frustrations, and inconsistent information. For example, patients and their family caregivers experienced long wait times for navigating the care process such as laboratory diagnostic tests and essential treatments such as radiation therapy, which caused disease progression and immense suffering. The frequent change of healthcare providers were mentioned as "the *constant change in doctors led to the need to explain my case repeatedly(P009),*" as well as issues with poor communication between inter- and intra-disciplinary teams leading to inefficiencies and emotional strain. One patient with metastatic cervical cancer said:

**Table 5. Summary of systemic barriers towards palliative care in Tikur Anbessa Specialized Hospital, Ethiopia.**

| Systemic barriers | Patients | Family caregivers |
|---|---|---|
| Financial burden/ challenges | "I travelled long distance after getting help from people who know and don't know me alongside God. I am so tired of everything." (P005) | "Our family struggles financially, sometimes relying on overseas assistance." (F/C001) |
| Availability and affordability of effective medicines | "It is difficult in obtaining certain medications, particularly pain relievers, extremely expensive in the private." (P011) "More effective pain relief options should be available…" (P002) | "There is drug called morphine, mostly not available, if available it is too expensive, 6000 birr for 30 days [~USD55.5]." (F/C003) |
| Access to diagnostic tests | "The CT scan, MRI and other tests in the hospital are not always inaccessible." (P002) | "…frustrated with the lack of common diagnostic tests in the hospital." (F/C007) |
| Inaccessibility of the holistic care | "Psychotherapy or religious counseling are unavailable in the hospital, as a result received prayers from family members via cellphone." (P003) | "The care is only available here, and it's a burden for the family—especially since private care is so expensive." – F/C004 |
| Constant change of health-care providers | "It is difficult always and frustrating to explain my case repeatedly due to the constant change in doctors." (P009) | --- |
| Lack of patient/family assistance | "Particularly during initial visit and then after navigating entire care is difficult." (P001) | "The environment is so complex, distressing." (F/C002) |
| High patient load and long waiting time | "Due to high patient load in the waiting area, usually it is difficult to have adequate time with doctor." (P006) | "The waiting area is so congested with lots of patients and their families." (F/C001) |
| Hospital infrastructure | "It is difficult to discuss some sensitive issues with the doctor. The rooms are uncomfortable." (P003) | "The hospital lacks basic facilities such as functioning toilets, and we have to walk over piled boxes to access them and difficult to use." (F/C 006) |

"A pharmacist told and concerned that the prescribed medication for my pain is wrong and I was confused and delays in getting my medication." (P009)

Furthermore, patients and family caregivers experienced difficulties with follow-up care, as access to healthcare providers outside of scheduled appointments is limited, and they must wait for specific days to be seen. One caregiver mentioned: "only Tuesdays and Thursdays are dedicated [to] palliative care." Additionally, the lack of communication and coordination between various healthcare facilities and practitioners added to the challenges faced in receiving comprehensive care. One patient said "[I did] not receive clear guidance and adequate information from the hospital we [were] referred from". Difficulties in accessing medications, diagnostics, and appropriate treatments highlighted gaps in healthcare system integration, further contributed to delays in care.

**Patient-centered holistic care**

Most of the key informants highlighted that palliative care in the hospital is not holistic and fully patient-centered. Multiple patients and family caregivers report experiencing significant gaps in addressing not just their physical needs but also emotional, psychological, and spiritual well-being. One patient who is on palliative care for about 3 months said:

"After a long waiting time the doctor asked me a few questions, [and] only prescribed some pain medicines. That is it." (P002)

Another patient added:

"Why did GOD put me in suffering? Since there is spiritual care in the hospital, I turn to my "father of repentance"/ "የንስሀ አባት" and feel somewhat better." (P008)

Financial barriers, poor service coordination, poor home-care integration, inadequate pain management, and poor hospital conditions all contribute to care that falls short of meeting patients' physical, emotional, and financial needs.

## Summary of suggested improvements

Suggested improvements from all participants, patients and family caregivers were to enhance access to medications, especially opioid analgesics, which are often expensive and hard to obtain. There is a need for more timely and consistent treatments, including radiation therapy, to prevent delays that can worsen conditions. The need for improved hospital conditions, such as cleanliness and functioning facilities, were also highlighted. Participants emphasized the importance of having better coordination in care, including more consistent communication and having a single point of contact for medical updates. Additionally, addressing financial barriers to treatment, particularly for diagnostics and treatment modalities, was a key concern. Many caregivers also suggested improving the emotional and psychological support for patients, as psychological support and counseling services were often absent. Lastly, increasing the availability of home-care services was recommended to alleviate some of the physical and financial strain on families.

## Triangulation of qualitative and quantitative data

Triangulation of quantitative and qualitative findings showed strong convergence for pain severity and place of residence, where lower satisfaction towards palliative care aligned with narratives of poor pain control and access barriers. Qualitative findings further expanded the quantitative results by identifying continuity of care, infrastructure, financial burden, and lack of holistic support as key contributors to patient distress that were not captured in the regression model (Table 6).

## Discussion

The findings from this study at TASH in Ethiopia reveal a significant level of patient with cancer dissatisfaction with hospital-based palliative care services, with an overall satisfaction rate of only 56.7%. This is notably lower compared to 76.8% in Japan [15], 88.2% in Bangladesh [14] for integrated home-based palliative care, and 93%–96% in the United States [24]. Higher satisfaction rates in other countries are often linked to better resource availability, comprehensive symptom management, and effective communication [7,8,11]. These findings highlight the critical need for resource strengthening and system-level improvements in Ethiopia's palliative care services.

Lower satisfaction rate in this study was significantly associated with greater pain severity, type of cancer and living outside of the city where treatment center is located. The association with greater pain severity is consistent with the understanding that uncontrolled pain can negatively impact patients' overall experience and satisfaction with care, underscoring the importance of effective pain management in palliative settings [10,14,15,25]. The type of cancer may influence satisfaction due to differences in disease trajectory, symptom burden, and treatment complexity such as in gynecological and gastrointestinal malignancies as compared to breast cancer [26]. Patients living outside the city may face additional barriers such as travel difficulties, limited access to consistent care, and financial strain, which can contribute to lower satisfaction. Employment status may reflect socioeconomic factors influencing patients' expectations and access to supportive resources. The lower satisfaction rate was further supported by qualitative findings, where patients and family caregivers highlighted issues like unavailability of pain-relieving medications (e.g., morphine), poor coordination and continuity of care, financial challenges, poor infrastructure, and physical inaccessibility of care (e.g., limited home-based palliative care where patients had to travel >600 km), which contributed to their dissatisfaction. Our findings are in line with multiple other studies that also indicated these systemic barriers remained a significant issue in LMICs, particularly in Ethiopia [6,7,9–11]. Conversely, factors such as performance status, employment status and disease relapse/recurrence were not significantly associated with satisfaction in our study. This may be due to, variability in patient perceptions, or the multidimensional nature of satisfaction that extends beyond clinical or employment status. Additionally, the cross-sectional design limits causal inference, and satisfaction may be influenced by unmeasured psychosocial or systemic factors.

**Global Public Health**
**PLOS**

**Table 6. Triangulation of quantitative and qualitative findings on factors influencing overall symptom distress and palliative care experience at TASH.**

| Key domain/ factor | Quantitative findings (MSAS total symptom distress) | Qualitative themes | Illustrative quotes (patients/family caregivers) | Integrated interpretation |
|---|---|---|---|---|
| **Place of residence** | Patients living outside Addis Ababa had significantly lower satisfaction ($\beta=-2.77$; 95% CI: $-5.29, -0.25$; $p=0.031$) | Inaccessibility of services; travel burden | "I travelled long distance after getting help from people who know and don't know me… I am so tired of everything." (P005) | Quantitative evidence of lower satisfaction among non-Addis residents is explained by long travel distances, financial strain, and limited access to timely care. |
| **Employment/ financial status** | Employed patients tended to have lower satisfaction, though not statistically significant ($\beta=-1.99$; $p=0.123$) | Financial burden and affordability challenges | "Our family struggles financially, sometimes relying on overseas assistance." (F/C001) | Although not statistically significant, qualitative data strongly indicate financial hardship as a major contributor to distress and dissatisfaction as a result of conflicts with work responsibilities, leading to income loss, job insecurity, and added financial stress for employed patients in a largely out-of-pocket health systems like Ethiopia. |
| **Cancer site** | Several cancer sites showed significantly lower satisfaction compared with breast cancer (e.g., CNS $\beta=-17.83$; $p=0.018$; Head & neck $\beta=-11.56$; $p<0.001$) | Complex symptoms and unmet needs | "After a long waiting time the doctor asked me a few questions and only prescribed pain medicines." (P002) | Higher distress in certain cancer sites reflects complex symptom profiles inadequately addressed by current palliative care services. Breast cancer treatment is one of the focused diseases for management, accessible and not complex like others. |
| **Pain severity** | Moderate pain ($\beta=-7.34$; $p<0.001$) and severe pain ($\beta=-11.5$; $p<0.001$) were strongly associated with higher symptom distress | Poor pain control; limited access to opioids | "Morphine is mostly not available, and if available it is too expensive." (F/ C003) | Strong statistical association between pain severity and satisfaction is reinforced by qualitative reports of inadequate pain management and medicine shortages. |
| **Continuity of care** | Not directly captured quantitatively, but indirectly reflected through satisfaction | Frequent provider changes; poor coordination | "The constant change in doctors led to the need to explain my case repeatedly." (P009) | Qualitative findings identify care discontinuity as a key contributor to distress, complementing quantitative results not measuring this construct. |
| **Access to care outside appointments** | Not directly captured quantitatively, but indirectly reflected through satisfaction | Restricted access and rigid scheduling | "It is difficult to seek treatment outside the scheduled appointments." (P008, F/C003) | Limited flexibility in care access exacerbates symptom burden, particularly for patients with severe pain or advanced disease. |
| **Hospital infrastructure** | Not directly captured quantitatively, but indirectly reflected through satisfaction | Poor sanitation and physical environment | "The hospital lacks basic amenities such as a functioning toilet." (F/ C006) | Poor infrastructure contributes to negative care experience and distress beyond clinical symptom severity. |
| **Holistic and patient-centered care** | Not directly captured quantitatively, but indirectly reflected through satisfaction | Lack of psychological, spiritual, and social support | "Psychotherapy or religious counseling are unavailable… I receive prayers by phone." (P003) | Quantitative distress scores likely underestimate the multidimensional suffering described in qualitative accounts. |

Pain management emerged as a critical issue in both quantitative and qualitative findings among patients with cancer, highlighting a significant gap in the provision of effective palliative care. Despite 17.7% of patients reporting severe pain, only a small number were able to access morphine. This discrepancy severely impacted overall patient dissatisfaction, as those experiencing moderate (52.6%) to severe pain (48.5%) reported lower levels of satisfaction as compared to those with ≤mild pain (61.2%). In interviews, patients and caregivers expressed frustration over the lack of availability and affordability of pain-relieving medications, with some indicating that such medications were often either stocked out or prohibitively expensive. This is further supported by evidence showing that access to opioid analgesics is a problem in LMICs

where only 1% of the poorest 50% of the global population can access and consume opioid analgesics, compared to the wealthiest 10% who consume 90% of the world's opioid analgesics [6,25,27].

In this study, patient satisfaction with palliative care services significantly differed by the type and location of cancer. Patients with reproductive, head and neck, musculoskeletal, lung, and central nervous system cancers reported notably lower satisfaction scores compared to those with breast cancer. This discrepancy may be due to the nature and complexity of these cancers, as well as challenges in accessing appropriate care for these cancers. These patients often require a combination of treatments, including surgery, radiation, and chemotherapy, and frequently face delays and unmet needs. Key informant interviews highlighted treatment delays as a frequent issue. For example, in Ethiopia, breast cancer treatment is available at over 24 government hospitals and more than 10 private facilities, with the government actively working to make breast cancer care more accessible "*close to homes*". In contrast, fewer hospitals provide care for other cancer types, which may impact treatment accessibility and contribute to lower satisfaction [28–30].

Moreover, overall satisfaction with palliative care services also differed by residence. Those residing outside of the city where the treatment center, TASH, is located, reported lower satisfaction scores compared to those living within the city. This disparity may be attributed to several interrelated challenges. Patients from outside city often face long travel distances, leading to higher transportation costs and increased physical strain. They may also incur extra expenses for accommodation and food, intensifying the financial burden. These challenges make it difficult to maintain regular follow-up visits, negatively affecting care continuity and perceived quality. In contrast, those from the city, incur fewer travel-related costs, and can better balance appointments with work and family responsibilities. Additionally, financial burden was a recurring theme in qualitative interviews, with many patients and caregivers relying on external financial support to cover medications and diagnostic costs [6,9,31].

The systemic barriers identified in both the qualitative and quantitative data at TASH highlight the urgent need to improve access to diagnostics and essential medications such as morphine. In contrast, the main source of dissatisfaction in the Japanese study was inadequate communication about prognosis and limited involvement of family members in care decisions [15]. Meanwhile, in Bangladesh, dissatisfaction was more related to the management of physical symptoms, especially pain, where access to strong pain relief like morphine is crucial [14]. These differences highlight the importance of addressing both structural and contextual factors in palliative care.

Respondents also emphasized the importance of enhancing infrastructure, and ensuring basic amenities such as comfortable patient rooms. Additionally, there is a need for more coordinated care, such as having a single point of contact, to support *"whole patient care"* by physical, emotional, social, and spiritual needs through better communication and continuity. Expanding financial assistance programs and integrating home-care services could ease the economic burden on patients and families and access to quality palliative care. Achieving this requires strong policy to establish a more supportive and holistic palliative care system in Ethiopia, comparable to the services models offered in Kenya and Uganda [32,33] and enhance patient and caregiver satisfaction.

This study has notable strengths. First, the use of a large sample size (n = 844) enhances the statistical reliability and generalizability of the findings. The mixed-methods design allowed for a more nuanced understanding of patient satisfaction that reflect personal and contextual experiences. Moreover, this study addresses a significant gap in the Ethiopian context, being one of the first to explore patient satisfaction with palliative care services directly from patients' perspectives.

However, the study also has limitations. Although the SATMED-Q is a validated and psychometrically sound instrument, and its domains; are highly relevant to evaluating palliative treatment experiences, it was not originally developed for palliative care, and may not fully capture the unique psychosocial and emotional dimensions specific to palliative care, such as spiritual concerns or existential distress. Furthermore, the findings are context-specific and may not be generalizable to other countries or healthcare systems with different palliative care structures. Use of purposive sampling for the qualitative and consecutive sampling technique had the potential for selection bias in participant recruitment and influenced the

representativeness of the study population and, consequently, the generalizability of the findings. The reliance on self-reported data introduces the possibility of recall or social desirability bias, particularly in a cultural setting where expressing dissatisfaction may be influenced by religious or societal norms. Lastly, caregiver satisfaction was not measured quantitatively, and future studies should consider structured assessment of both patients' and caregivers' satisfaction to better capture differences in experiences and expectations.

## Conclusion

The quantitative and qualitative findings reveal key areas in need of palliative care service improvement for patients diagnosed and treated for cancer in TASH and Ethiopia at large. Findings revealed that pain management remains a major challenge due to limited access to essential medications such as morphine and a major reason for dissatisfaction. Additionally, the lack of continuity and coordination in care and poor infrastructure were mentioned as systemic barriers and emphasized the need for systemic reforms to offer more patient-centered and effective palliative care services in Ethiopia.

## Supporting information

**S1 Appendix. Semi-structured interview guide.**
(DOCX)

**S2 Appendix. Validity and reliability test results.**
(DOCX)

## Acknowledgments

We would like to thank Prof. Anteneh Belete, Dr. Eskinder Eshetu, Girma Tekle, Muluken Nigatu, and Tesfa Marew for their project management contribution to this study. We would also like to thank all the study participants for their willingness and time to participate in the study.

## Author contributions

**Conceptualization:** Atalay Mulu Fentie, Edom Seife, Sachiko Ozawa.

**Data curation:** Atalay Mulu Fentie, Edom Seife, Sachiko Ozawa, Teferi Gedif Fenta.

**Formal analysis:** Atalay Mulu Fentie.

**Methodology:** Atalay Mulu Fentie, Edom Seife, Sachiko Ozawa, Teferi Gedif Fenta.

**Project administration:** Edom Seife, Sachiko Ozawa, Teferi Gedif Fenta.

**Software:** Atalay Mulu Fentie.

**Supervision:** Edom Seife, Sachiko Ozawa, Teferi Gedif Fenta.

**Validation:** Atalay Mulu Fentie, Edom Seife, Sachiko Ozawa, Teferi Gedif Fenta.

**Visualization:** Atalay Mulu Fentie, Edom Seife, Sachiko Ozawa, Teferi Gedif Fenta.

**Writing – original draft:** Atalay Mulu Fentie.

**Writing – review & editing:** Atalay Mulu Fentie, Edom Seife, Sachiko Ozawa, Teferi Gedif Fenta.

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
