## [Decision Letter · Decision Letter 0]

17 Nov 2025

PGPH-D-25-02751

Experience and satisfaction towards palliative care in an Ethiopian tertiary care setting: A mixed methods study of patients with cancer and caregivers

Dear Dr. Fentie,

Thank you for submitting your manuscript to PLOS Global Public Health. After careful consideration, we feel that it has merit but does not fully meet PLOS Global Public Health’s publication criteria as it currently stands. Therefore, we invite you to submit a revised version of the manuscript that addresses the points raised during the review process.

Addressing the comments from the reviewers will improve the quality of the paper. Though most of the comments require minor changes, the major modification is needed in the way the results are presented. This being a mixed-methods research, it is ideal to integrate the results instead of presenting them separately.

We look forward to receiving your revised manuscript.

Kind regards,

Sonali Sarkar

Academic Editor

Journal Requirements:

1. Please send a completed 'Competing Interests' statement, including any COIs declared by your co-authors. If you have no competing interests to declare, please state "The authors have declared that no competing interests exist". Otherwise please declare all competing interests beginning with the statement "I have read the journal's policy and the authors of this manuscript have the following competing interests:"

1. Please clarify all sources of funding (financial or material support) for your study. List the grants (with grant number) or organizations (with url) that supported your study, including funding received from your institution.

2. State the initials, alongside each funding source, of each author to receive each grant.

3. State what role the funders took in the study. If the funders had no role in your study, please state: “The funders had no role in study design, data collection and analysis, decision to publish, or preparation of the manuscript.”

4. If any authors received a salary from any of your funders, please state which authors and which funders.

3. Please provide separate figure files in .tif or .eps format.

Additional Editor Comments:

This is a good study and on a very pertinent topic of palliative care for cancer patients in a LMIC. Though the paper is well written, this being a mixed-methods research, integration of the quantitative and qualitative results should be done.

Reviewers' comments:

Reviewer's Responses to Questions

**Comments to the Author**

1. Does this manuscript meet PLOS Global Public Health’s publication criteria? Is the manuscript technically sound, and do the data support the conclusions? The manuscript must describe methodologically and ethically rigorous research with conclusions that are appropriately drawn based on the data presented.? Is the manuscript technically sound, and do the data support the conclusions? The manuscript must describe methodologically and ethically rigorous research with conclusions that are appropriately drawn based on the data presented.? Is the manuscript technically sound, and do the data support the conclusions? The manuscript must describe methodologically and ethically rigorous research with conclusions that are appropriately drawn based on the data presented.? Is the manuscript technically sound, and do the data support the conclusions? The manuscript must describe methodologically and ethically rigorous research with conclusions that are appropriately drawn based on the data presented.

Reviewer #1: Partly

Reviewer #2: Yes

Reviewer #3: Yes

Reviewer #4: Yes

2. Has the statistical analysis been performed appropriately and rigorously?

Reviewer #1: Yes

Reviewer #2: Yes

Reviewer #3: Yes

Reviewer #4: Yes

3. Have the authors made all data underlying the findings in their manuscript fully available (please refer to the Data Availability Statement at the start of the manuscript PDF file)?

The PLOS Data policy requires authors to make all data underlying the findings described in their manuscript fully available without restriction, with rare exception. The data should be provided as part of the manuscript or its supporting information, or deposited to a public repository. For example, in addition to summary statistics, the data points behind means, medians and variance measures should be available. If there are restrictions on publicly sharing data—e.g. participant privacy or use of data from a third party—those must be specified.requires authors to make all data underlying the findings described in their manuscript fully available without restriction, with rare exception. The data should be provided as part of the manuscript or its supporting information, or deposited to a public repository. For example, in addition to summary statistics, the data points behind means, medians and variance measures should be available. If there are restrictions on publicly sharing data—e.g. participant privacy or use of data from a third party—those must be specified.requires authors to make all data underlying the findings described in their manuscript fully available without restriction, with rare exception. The data should be provided as part of the manuscript or its supporting information, or deposited to a public repository. For example, in addition to summary statistics, the data points behind means, medians and variance measures should be available. If there are restrictions on publicly sharing data—e.g. participant privacy or use of data from a third party—those must be specified.requires authors to make all data underlying the findings described in their manuscript fully available without restriction, with rare exception. The data should be provided as part of the manuscript or its supporting information, or deposited to a public repository. For example, in addition to summary statistics, the data points behind means, medians and variance measures should be available. If there are restrictions on publicly sharing data—e.g. participant privacy or use of data from a third party—those must be specified.

Reviewer #1: Yes

Reviewer #2: Yes

Reviewer #3: Yes

Reviewer #4: Yes

4. Is the manuscript presented in an intelligible fashion and written in standard English?

Reviewer #1: Yes

Reviewer #2: No

Reviewer #3: Yes

Reviewer #4: Yes

5. Review Comments to the Author

Reviewer #1: Introduction: Line 88 has typographical error.

Line 117: Sample size calculation of quantitative part was unclear.

Table 1: Add N for Type of comorbidity

Patients residing outside of Addis Ababa is general statement. The characteristics of such classification like how much distance away from TASH, absence of tertiary hospitals in surrounding radius would make it clear to ascertain the satisfaction and to avoid confounders.

The details on palliative care services in the setting which was provided to the patients was not explained. Whether the patients who were given curative treatment provided with palliative care could be explained to understand the situation.

Line 262-269: The authors explanation of association with various cancers and pain is appreciated taking into consideration the scientific way of explain the multiple linear regression output but it looks more statistical and can be modified. “Every additional patient” doesn’t make sense in this study point and can be explained better.

The satisfaction scores in unemployed were better compared with employed and that could have been explored in qualitative. The qualitative part is written well but it needs some refinement to explain the quantitative results.

The Amharic text font is not supported. Try to correct it and make it visible.

The biases involved in selection of participants could be mentioned.

Reviewer #2: 1. Details of the palliative care services which are currently being provided are missing. Another crucial detail is how many are aware of these services, how many are eligible for these services, and how many availed these services ? Then the satisfaction will be measured among those who all in need tried to utilise the services.

2. Details about how many patients approached, how many refused to participate. There would patients who could not comprehend the questionnaire due to the severity of illness or distress, as 10.9% of patients had the status of ‘couldn’t function’.

3. Satisfaction measured among patients was measured objectively, while the satisfaction of caregivers is missing, as both have a very different journey and expectations in the hospital, which affects the majority’s satisfaction.

4. While qualitative research methodology is used, triangulation is missing. Here, utilising a focus group discussion instead of in-depth interviews would have given a better generation of data.

5. Adding the perspective of healthcare providers and administration would have given a 360-degree picture of the situation. As patient satisfaction is just a tip of the iceberg.

6. Assumptions and references for sample size calculation are missing.

7. For assessing the factors associated with patient satisfaction, it needs to have proper exploration /defining of confounders, exposure, and mediators.

Reviewer #3: 1. It will be useful to describe any existing policy on Palliative care in Ethiopia and opioid availability status

2. a description about the multidisciplionary team providing palliative care and type of services and any home care provided to know the background

Reviewer #4: The manuscript deals with data on needs and satisfaction of palliative care patients, which if published could provide additional evidence to existing literature from a low income country like Ethiopia.

6. PLOS authors have the option to publish the peer review history of their article (what does this mean?). If published, this will include your full peer review and any attached files.). If published, this will include your full peer review and any attached files.). If published, this will include your full peer review and any attached files.). If published, this will include your full peer review and any attached files.

**Do you want your identity to be public for this peer review?** For information about this choice, including consent withdrawal, please see our Privacy Policy....

Reviewer #1: No

Reviewer #2: No

Reviewer #3: No

Reviewer #4: No

Figure Resubmissions:

After uploading your figures to PLOS’s NAAS tool - https://ngplosjournals.pagemajik.ai/artanalysis NAAS will process the files provided and display the results in the "Uploaded Files" section of the page as the processing is complete. If the uploaded figures meet our requirements (or NAAS is able to fix the files to meet our requirements), the figure will be marked as "fixed" above. If NAAS is unable to fix the files, a red "failed" label will appear above. When NAAS has confirmed that the figure files meet our requirements, please download the file via the download option, and include these NAAS processed figure files when submitting your revised manuscript.

---

## [Editor Report · Decision Letter 1]

30 Mar 2026

PGPH-D-25-02751R1

Experience and satisfaction towards palliative care in an Ethiopian tertiary care setting: A mixed methods study of patients with cancer and caregivers

Dear Dr. Fentie,

Thank you for modifying your manuscript to as per the reviewers' suggestions. After careful consideration, we feel that it has merit but does not fully meet PLOS Global Public Health’s publication criteria as it currently stands. The introduction, objective and conclusion need to be more specific as the study is conducted only among cancer patients. Language corrections are required at multiple places. Therefore, we invite you to submit a revised version of the manuscript that addresses the points raised during the review process.

As the corresponding author, your ORCID iD is verified in the submission system and will appear in the published article. PLOS supports the use of ORCID, and we encourage all coauthors to register for an ORCID iD and use it as well. Please encourage your coauthors to verify their ORCID iD within the submission system before final acceptance, as unverified ORCID iDs will not appear in the published article. *Only* the individual author can complete the verification step; PLOS staff the individual author can complete the verification step; PLOS staff the individual author can complete the verification step; PLOS staff the individual author can complete the verification step; PLOS staff *cannot* verify ORCID iDs on behalf of authors.verify ORCID iDs on behalf of authors.verify ORCID iDs on behalf of authors.verify ORCID iDs on behalf of authors.

We look forward to receiving your revised manuscript.

Kind regards,

Sonali Sarkar

Academic Editor

Journal Requirements:

Additional Editor Comments (if provided):

All the comments of the reviewers have been addressed satisfactorily. However, two points need to improved upon before the manuscript can be considered for publication.

1. The study is only on cancer patients. Therefore, the justification should be appropriately modified. The objectives and the conclusion should also be stated specifically mentioning the study population as cancer patients.

2. Language correction is needed throughout the manuscript.

Reviewers' comments:

 Figure Resubmissions:

After uploading your figures to PLOS’s NAAS tool - https://ngplosjournals.pagemajik.ai/artanalysis NAAS will process the files provided and display the results in the "Uploaded Files" section of the page as the processing is complete. If the uploaded figures meet our requirements (or NAAS is able to fix the files to meet our requirements), the figure will be marked as "fixed" above. If NAAS is unable to fix the files, a red "failed" label will appear above. When NAAS has confirmed that the figure files meet our requirements, please download the file via the download option, and include these NAAS processed figure files when submitting your revised manuscript.

---

## [Editor Report · Decision Letter 2]

7 Apr 2026

Experience and satisfaction towards palliative care in an Ethiopian tertiary care setting: A mixed methods study of patients with cancer and caregivers

PGPH-D-25-02751R2

Dear Dr. Fentie,

We are pleased to inform you that your manuscript 'Experience and satisfaction towards palliative care in an Ethiopian tertiary care setting: A mixed methods study of patients with cancer and caregivers' has been provisionally accepted for publication in PLOS Global Public Health.

Best regards,

Sonali Sarkar

Academic Editor